

# Effect of proprioceptive neuromuscular facilitation (PNF) technique on posture, balance and gait characteristics of older adults with scapular dyskinesis: a randomized controlled trial

Hasan Cellatoğlu[1,2] and Beliz Belgen Kaygısız[1]

[1] Department of Physiotherapy and Rehabilitation, European University of Lefke, Lefke, Mersin 10, Turkey
[2] Department of Physiotherapy and Rehabilitation, Cyprus Aydin University, Kyrenia, Mersin 10, Turkey

## ABSTRACT

**Background.** Proprioceptive neuromuscular facilitation (PNF) techniques are being utilized among rehabilitation protocols for older adults because they are based on proprioception and coordination activities. The main purpose of this study was to analyze whether including scapular PNF in an intervention improves the posture, balance, and gait characteristics of older adults with scapular dyskinesis.

**Methods.** Forty-eight older adults (26 females; 22 males; mean $\pm$ SD age, 69.17 $\pm$ 4.36) with scapular dyskinesis were randomized into two groups: PNF and conventional exercise group (PNF+Ex) ($n = 23$) and conventional exercise group (Ex) ($n = 25$). Both groups received treatment three times a week for eight weeks and each session lasted 60 min. Outcome measurements included the Lateral Scapular Slide Test (LSST), Dynamic Gait Index (DGI), Walking Speed Test, Timed Up and Go Test (TUG), Functional Reach Test (FRT), and PostureScreen Mobile (PSM) iOS application, which were collected at baseline, after treatment, and three months after the last session.

**Results.** Both groups showed improvements in scapular dyskinesis, gait characteristics, and posture. When the two groups were compared, the PNF+Ex group had significantly different LSST results in three positions (0°, 45°, and 90°) ($P < 0.000$, $F = 5.414$, $F = 11.440$ and $F = 31.234$) and postural results for head tilt ($F = 4.116$, $P = 0.045$), vertical displacement of head ($F = 5.790$, $P < 0.000$) and shoulder tilt ($F = 6.959$, $P < 0.008$). There were no statistically significant differences in the improvement in balance and gait characteristics between the groups ($P > 0.05$).

**Conclusion.** Scapular PNF can be added to the rehabilitation of older adults with scapular dyskinesis to improve their scapular disposition and posture. A longer treatment duration is suggested to improve the gait and balance.

# INTRODUCTION

Aging is a progressive process in which physiological and functional changes occur, and consequently, posture, balance, and gait-related problems are also expected to appear (*Alves*

Corresponding author
Hasan Cellatoğlu,
hasancellatoglu@cau.edu.tr

*et al., 2016*). Thoracic kyphosis is one of the most commonly observed postural problems and it is usually accompanied by scapular dyskinesis (*Otoshi et al., 2014*). As previous studies stated, scapular dyskinesis can alter thoracic spine alignment and mechanics as thoracic kyphosis has been found to be influenced by scapular dysfunction, indicating a broader impact on postural control (*Otoshi et al., 2014*; *Kawamata et al., 2024*; *Tedeschi et al., 2024*; *Telli & Sağlam, 2022*). Thoracic kyphosis itself has been associated with decreased mobility, reduced gait speed, and impaired balance. As it was stated by several studies, increased kyphosis can significantly affect these functional outcomes (*Eum et al., 2013*; *Cheng et al., 2023*; *Gasavi Nezhad, Gard & Arazpour, 2025*). Scapular functions provide postural control, shoulder stabilization, and control of the arm swing, which are crucial during normal gait, and disturbances in these functions lead to disturbances in normal gait functions (*Osborn & Homberger, 2015*; *Cools et al., 2013*). These findings collectively demonstrate the connection between scapular dyskinesis and thoracic changes, which in turn contribute to impairments in functional mobility and balance. Hence, scapular dyskinesis together with thoracic kyphosis leads to increased risk of falls, which is caused by disturbed balance, decreased functional mobility, and gait deviations in older adults (*Ardakani et al., 2022*; *Shiravi et al., 2019*). Therefore, since scapular misalignment may contribute to thoracic spinal misalignment, the inclusion of scapula oriented rehabilitation strategies is important, as supported by the literature (*Shiravi et al., 2019*).

As a result, for the treatment of scapular dyskinesis, proprioceptive neuromuscular facilitation (PNF) can be an effective intervention that improves motor control, muscle activation, muscle strength, proprioception, and stabilization (*Hwang, Lee & Lim, 2021*; *Balcı et al., 2016*).

PNF benefits from the body's cutaneous, proprioceptive, and auditory inputs to improve motor function and can be a vital element in rehabilitation (*Kumar & George, 2024*; *Hwang, Lee & Lim, 2021*). For older adults, recent studies demonstrated that including PNF in rehabilitation programs can effectively improve muscle activity, functional ability and balance (*Zaworski & Latosiewicz, 2021*; *Zaidi et al., 2023*; *Xiong et al., 2024*). These findings suggest that PNF interventions may play an important role in improving overall physical function in older adults. Previous studies by *Joshi, Shridhar & Jayaram (2020)* and *Hwang, Lee & Lim (2021)*, stated that application of scapular PNF interventions to office workers and adhesive capsulitis patients with scapular dyskinesis improved functional mobility and scapular position. Thus, with PNF, by enhancing thoracic muscle endurance where the scapula directly related, as impairments decrease, in total, improvement in the functional level can be observed (*Hwang, Lee & Lim, 2021*; *Uhl et al., 2009*). For difficult to correct postural misalignments such as thoracal kyphosis, it could be possible to decrease the impairments affected by kyphosis by enhancing scapular muscle activity. Consequently, with decreased impairments, improvements in posture, balance, scapular dyskinesis, and gait related outcomes are expected to be observed with scapular PNF (*Uhl et al., 2009*).

Although recent studies have shown associations between kyphosis-scapular dyskinesis and functional limitations (*Ardakani et al., 2022*; *Shiravi et al., 2019*), there is limited literature about whether adding scapular PNF to exercise programs can be an effective intervention for improving balance, gait, and posture. The main purpose of this study

was to analyze whether scapular PNF combined with a conventional exercise program improves the posture, balance, and gait characteristics of older adults with scapular dyskinesis compared to conventional exercise programs. We hypothesized that exercise combined with scapular PNF intervention would be more effective in improving posture, balance, and gait characteristics in older adults with scapular dyskinesis than conventional exercise.

## MATERIALS & METHODS

### Study design

This randomized controlled trial with parallel intervention groups was carried out after ethical approval was obtained from the University Ethics Committee of the University of Lefke (BAYEK022.04) on April 3, 2023, and registered at http://www.clinicaltrials.gov (NCT05893303) on May 30, 2023.

### Participants

Participants were recruited from Guzelyurt and Nicosia in North Cyprus, and contacted by contacting the related municipality. Fifty-five participants were assessed for enrolment, and five were excluded. After randomization, two of the participants were discontinued. The randomization procedure is illustrated in Fig. 1. Therefore, older adults (26 females; 22 males; mean $\pm$ SD age, 69.17 $\pm$ 4.36) with scapular dyskinesis (positive Visual Scapular Dyskinesis Test and Lateral Scapular Slide Test results) who were able to walk without any assistive devices were included in study. The exclusion criteria were severe neurological problems or impairments that caused issues in applying the intervention and undergoing physiotherapy during the past three months. All participants signed an informed consent form before enrolment. Data were collected at the Fizyomorfo Physiotherapy and Rehabilitation Center between May 2023 and May 2024.

### Power analysis and randomization

Because of consisting of similar population and outcomes, the sample size was determined based on a study evaluating balance and mobility outcomes in elderly individuals (*Mesquita et al., 2015*). In the study "Effects of two exercise protocols on postural balance of elderly women: a randomized controlled trial" it was determined that the effect size for balance was $F = 0.77$ (*Mesquita et al., 2015*). Accordingly, assuming that the effect size was high in the study, the sample size required for 95% ($1-\alpha = 0.95$) power at $f = 0.50$, and $\beta = 0.05$ levels was calculated as 38 participants using G*Power 3.1.9.2. Six dropouts were added to each group, and the total number of participants was calculated as 50. Participants were randomized into two groups: PNF combined with conventional exercise (PNF+Ex) ($n = 25$) or conventional exercise (Ex) ($n = 25$), using GraphPad software and the block randomization technique. A statistician generated the sequence and sent it to the project manager. The allocation sequence was kept concealed from the researchers, and participant screening was conducted using sealed envelopes. Participants were not informed of the group to which they were allocated. Although both treatment and measurements were conducted by the same therapist, objective and standardized measurement tools that did

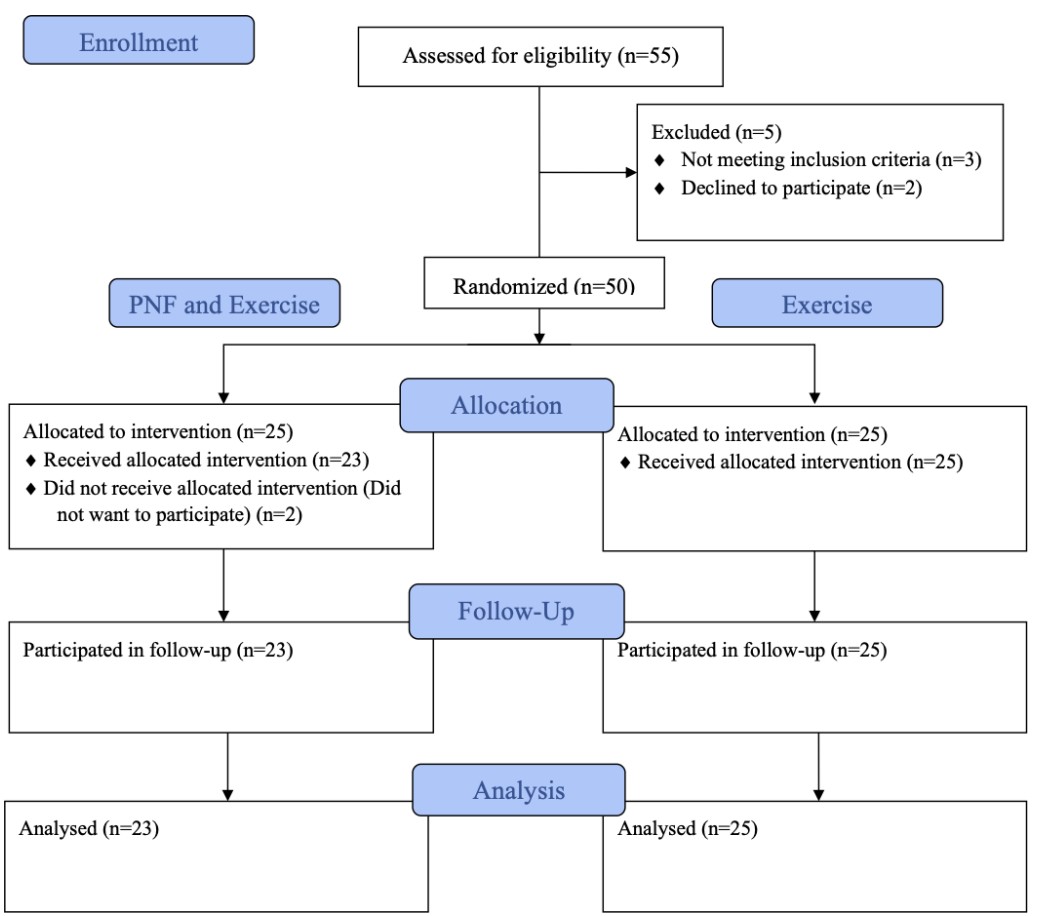

**Figure 1 Flow diagram of participant recruitment, randomization, group allocation and analysis processes.**

not require subjective interpretation were selected to minimize the risk of measurement bias.

## Intervention

Both groups underwent interventions three times per week for eight weeks, and each session lasted 60 min. For the PNF+Ex group, the sessions included scapular PNF interventions, including rhythmic initiation and repeated contraction-facilitation techniques for all scapular patterns (anterior elevation and posterior depression—posterior elevation and anterior depression). In addition to PNF interventions, conventional exercise programs consisting of strengthening exercises, gait, and balance training were applied. This program included strengthening exercises for the back and abdominal muscles, standing activities in a closed position and in tandem position, standing on an uneven surface, reciprocal movements of extremities, and stance and swing phase of the gait, which included stepping forward and sideways, weight transfer, and gait education (walking three lapses within two m parallel bar). For the Ex group, the sessions included the same conventional exercise program as the PNF+Ex group. Between exercises, 20 s resting periods were given to

participants to prevent fatigue and enhance effects of PNF (*Kaur, 2017*). Adherence to all scheduled sessions was ≥95%, with no significant difference between the groups. Details of the exercise and PNF protocols, including specific exercises, intensity, session structure and progression strategies for both groups, are provided in the (File S1).

## Outcome measurements

Measurements were taken at baseline, after the last session, and after three months of follow-up. Sociodemographic characteristics such as sex, age, body mass index (BMI), background, family history, smoking and alcohol habits, education status, marital status, physiotherapy history, and exercise habits were recorded at baseline in a dedicated form.

### Scapular dyskinesis

To detect the presence of scapular dyskinesis, the Visual Scapular Dyskinesis Test (VSDT) was performed. The test consisted of shoulder elevation in the sagittal plane while observing whether there was any prominence at the medial edge of the scapula (*Uhl et al., 2009*). Subsequently, the Lateral Scapular Slide Test (LSST) was applied to determine the severity and level of scapular dyskinesis. The test consisted of positioning the upper extremities in three different positions (0° abduction, 45° abduction, 90° abduction). At each position, the distance between the spinous process and lower angle of the scapula was measured using a measurement tape. If the difference between measurements in the same position was more than 1.5 centimeters (cm), it was recorded as scapular dyskinesis (*Kibler, 1998*).

### Posture

Posture measurements were performed utilizing the PostureScreen Mobile Application (PSM) (PostureCo Inc.). Photographs were taken by the researcher using the app's built-in camera, which provided on-screen instructions to capture standardized anterior and lateral views. All images were captured during the assessments and analyzed directly within the app. Measurements were obtained from these images, and reference points (*e.g.*, ears, shoulders, C5 and T5 vertebrae) were labeled. After the procedure, data were extracted and interpreted for shifts (cm), tilts (degrees), and vertical displacements (degrees) of the head, shoulders, and thorax. As the obtained values approached zero, the results were considered improved. According to *Boland et al. (2016)*, PSM is a validated and reliable tool for postural assessment.

### Functional mobility

Functional mobility was measured using the Timed Up and Go (TUG) test. TUG test results for functional mobility. During this test, after 'Go,' command participants stood up from a normal chair, walked three m, turned, walked back to the chair and sat. The time began with the command 'Go' and ended when the participants sat back. Data in this test were obtained in time (seconds), repeated three times, and the shortest performance time was recorded (*Podsiadlo & Richardson, 1991*).

### Dynamic balance

Dynamic balance measurements were performed using the Functional Reach Test (FRT). Participants stood upright next to a wall with their arm extended forward, parallel to the

floor, with the hand in a fist position. The starting position was marked at the tip of the third metacarpal. Participants were asked to reach forward as far as possible without moving their feet. The distance (cm) between the starting and ending positions were recorded. The test was repeated three times and the average of the three results was taken for analysis. As the obtained distance increased, dynamic balance was considered improved (*Ferreira, Raimundo & Marmeleira, 2021*).

### Gait characteristics

Gait characteristics were assessed using the Dynamic Gait Index (DGI), which evaluates walking ability under various conditions. The test includes eight tasks (walking at normal speed, changing walking speed, walking with head turns, pivot turning, stepping over and around obstacles, walking with a narrow base, and stair climbing). Each task was scored from 0 to 3, with a total maximum score of 24. Scores were interpreted as follows: disturbed balance if the result was <19, and safe ambulation if the result was >22 (*Herman et al., 2009*). Turkish validation of the test was conducted by *Zirek et al. (2023)*.

### Walking speed

Walking speed results were obtained by asking participants to walk a total of eight m straight distance. Measurements in seconds were obtained at a marked five m distance. The results were calculated in meters per second (m/s) (*Fritz & Lusardi, 2009*).

## Statistical analysis

The Statistical Package for Social Sciences (SPSS 29.0) was used for statistical analysis. All outcomes are expressed as the arithmetic average $\pm$ standard deviation (X $\pm$ SD). The chi-square test was used to compare the sociodemographic characteristics of the participants according to their groups. Skewness–kurtosis coefficients and Shapiro–Wilk tests were applied to the normal distribution of the data. Since the research data conformed to a normal distribution, 2-way repeated measurements analysis of variance (ANOVA) was applied to compare the participants' walking speed, TUG, DGI, FRT, LSST, and PSM. Statistical significance was set at $p \leq 0.05$.

## RESULTS

### Sociodemographic characteristics

Forty-eight older adults (26 females; 22 males; mean $\pm$ SD age, 69.17 $\pm$ 4.36) with scapular dyskinesis were included in the study and they were randomly divided into two groups. No statistically significant differences were found in the sociodemographic characteristics of participants. Table 1 presents the participants' sociodemographic data.

### Analysis of intervention effects

Before treatment, after treatment, and follow-up comparisons of the statistical results of scapular dyskinesis, posture, functional mobility, dynamic balance, gait, and walking speed results are listed.

For scapular dyskinesis results, there were significant differences in the LSST results between the groups. The decrease was higher in each position for the PNF+Ex group: 0°,

**Table 1 Baseline sociodemographic data of participants.**

| Variable[a] | PNF and exercise (n = 23) | Exercise (n = 25) | Total (n = 48) |
|---|---|---|---|
| Sex | | | |
| Female | 11 (47.83) | 15 (60) | 26 (54.17) |
| Male | 12 (52.17) | 10 (40) | 22 (45.83) |
| Age (years), mean ± SD | 68.48 ± 3.38 | 69.80 ± 5.08 | 69.17 ± 4.36 |
| BMI (kg/m$^2$), mean ± SD | 24.89 ± 3.30 | 28.82 ± 4.10 | 26.94 ± 4.20 |
| Educational background | | | |
| Primary | 4 (17.39) | 8 (32) | 12 (25) |
| High school | 8 (34.78) | 12 (48) | 20 (41.67) |
| University | 11 (47.83) | 5 (20) | 16 (33.33) |
| Marital status | | | |
| Married | 18 (78.26) | 19 (76) | 37 (77.08) |
| Single | … | 1 (4) | 1 (2.08) |
| Divorced | 5 (21.74) | 5 (20) | 10 (20.83) |
| Employment status | | | |
| Non-working | 4 (17.39) | 7 (28) | 11 (22.92) |
| Retired | 19 (82.61) | 18 (72) | 37 (77.08) |
| Number of chronic diseases | | | |
| No | 9 (39.13) | 10 (40) | 19 (39.58) |
| 1 | 5 (21.74) | 8 (32) | 13 (27.08) |
| >1 | 9 (39.13) | 7 (28) | 16 (33.33) |
| Number of family history of chronic disease | | | |
| No | 10 (43.48) | 7 (28) | 17 (35.42) |
| 1 | 5 (21.74) | 5 (20) | 10 (20.83) |
| >1 | 8 (34.78) | 13 (52) | 21 (43.75) |
| Number of medication | | | |
| No | 14 (60.87) | 15 (60) | 29 (60.42) |
| 1 | 3 (13.04) | 5 (20) | 8 (16.67) |
| >1 | 6 (26.09) | 5 (20) | 11 (22.92) |
| Alcohol habit (yes) | 7 (30.43) | 9 (36) | 16 (33.33) |
| Smoking habit (yes) | 9 (39.13) | 10 (40) | 19 (39.58) |
| Exercise habit (yes) | 4 (17.39) | 5 (20) | 9 (18.75) |
| Type of exercise | | | |
| No exercise habit | 19 (82.61) | 20 (80) | 39 (81.25) |
| Walking | 4 (17.39) | 3 (12) | 7 (14.58) |
| Pilates | … | 1 (4) | 1 (2.08) |
| Walking and Pilates | … | 1 (4) | 1 (2.08) |
| Receiving physiotherapy (yes)[b] | 4 (17.39) | 5 (20) | 9 (18.75) |

**Notes.**

Abbreviations: PNF, proprioceptive neuromuscular facilitation; BMI, body mass index.

[a] Values are n(%) unless otherwise indicated.

[b] Physiotherapy was not received 3 months before participation in the study.
**Table 2  Comparison of lateral scapular slide test differences results.**

| Outcome[a] (cm) | Group | Before treatment | After treatment | Follow up | F | P value | $\eta^2$ |
|---|---|---|---|---|---|---|---|
| 0° Abd Interscapular Difference | PNF and Ex | 1.63 ± 0.68 | 0.94 ± 0.68 | 1 ± 0.39 | T: 31.723 | 0.000[b] | 0.506 |
| | Ex | 2.12 ± 0.62 | 1.64 ± 0.57 | 1.78 ± 0.58 | T*G: 5.414 | 0.007[b] | 0.149 |
| 45° Abd Interscapular Difference | PNF and Ex | 2 ± 0.76 | 0.78 ± 0.57 | 0.86 ± 0.61 | T: 35.581 | 0.000[b] | 0.551 |
| | Ex | 2.08 ± 0.86 | 1.7 ± 0.73 | 1.9 ± 0.81 | T*G: 11.440 | 0.000[b] | 0.283 |
| 90° Abd Interscapular Difference | PNF and Ex | 2.07 ± 1.03 | 0.94 ± 0.53 | 0.96 ± 0.52 | T: 76.048 | 0.000[b] | 0.724 |
| | Ex | 2.19 ± 0.67 | 1.65 ± 0.61 | 1.91 ± 0.49 | T*G: 31.234 | 0.000[b] | 0.519 |

Notes.

Abbreviations: cm, centimetres; F, variance of the group means divided by mean of the within group variances; $\eta^2$, measure of effect size; Abd, abduction; PNF, proprioceptive neuromuscular facilitation; Ex, exercise; T, time; G, group.

[a]Values are mean ± SD.

[b]$p < 0.05$.

45°, and 90° ($p < 0.000$, $F = 5.414$, $F = 11.440$ and $F = 31.234$). In addition, follow-up results at each abduction position in the PNF+Ex group showed more improvement than the follow-up results of the Ex group. The LSST results are listed in Table 2.

According to the posture results, before treatment, after treatment, and follow-up results, improvements were observed in both the PNF+Ex and Ex groups. There were statistically significant differences between the groups in three items: lateral view vertical displacement of the head ($F = 5.790$, $p < 0.000$), anterior view shoulder tilt ($F = 6.959$, $p = 0.008$), and lateral view vertical displacement of the shoulder ($F = 59.201$, $p < 0.000$) in favor of the PNF+Ex group when compared to before treatment, after treatment, and follow up results. There were no statistically significant differences in other items between the groups ($p > 0.05$). Postural results are presented in Table 3.

When balance and gait characteristics were analyzed, there were significant improvements in both groups, although no significant differences between the groups were found for TUG, FRT, DGI and walking speed comparisons ($p > 0.05$). For DGI, although there was no significant difference, the results showed greater improvement in the PNF+Ex group. In addition, for FRT, the PNF+Ex results showed more consistency during the follow-up results. The results are summarized in Table 4.

## DISCUSSION

This study was conducted to observe the effects of scapular PNF interventions on the posture, balance, and gait characteristics of older adults with scapular dyskinesis. To the best of our knowledge, this is the first randomized controlled study to observe the effects of scapular PNF interventions on gait and balance related functional outcomes in scapular dyskinesis. The most clinically relevant finding of the study was improvements that were more notable regarding scapular dyskinesis and postural improvements in the PNF+Ex group. When comparing the two groups, improvements were observed, although no statistically significant differences were found, and improvements in balance, gait, and walking speed were observed.

**Table 3 Comparison of posture differences results.**

| Outcome[a] | Group | Before treatment | After treatment | Follow up | F | p | $\eta^2$ |
|---|---|---|---|---|---|---|---|
| Anterior View Head Shift (cm) | PNF and Ex | −0.12 ± 1.13 | 0.06 ± 1.34 | −0.01 ± 1.36 | T: 0.84 | 0.404 | 0.018 |
| | Ex | 0.05 ± 1.62 | −0.03 ± 1.5 | −0.27 ± 1.48 | T*G: 1.622 | 0.203 | 0.034 |
| Anterior View Head Tilt (cm) | PNF and Ex | −1.73 ± 2.69 | −1.1 ± 2.42 | −1.19 ± 2.35 | T: 4.116 | 0.045[b] | 0.1 |
| | Ex | −0.75 ± 2.85 | −0.67 ± 2.58 | −0.62 ± 2.56 | T*G: 3.068 | 0.083 | 0.077 |
| Lateral View Head Shift (cm) | PNF and Ex | 2.87 ± 1.07 | 3.28 ± 1.44 | 3.26 ± 1.47 | T: 0.007 | 0.944 | 0 |
| | Ex | 4.44 ± 2.46 | 4.06 ± 2.13 | 4.05 ± 2.20 | T*G: 1.267 | 0.224 | 0.103 |
| Lateral View Vertical Displacement of Head (degrees) | PNF and Ex | 9.2 ± 3.53 | 5.88 ± 2.72 | 6.45 ± 3.08 | T: 197.528 | 0.000[b] | 0.811 |
| | Ex | 10.2 ± 4.54 | 9.63 ± 4.53 | 9.96 ± 4.7 | T*G: 111.254 | 0.000[b] | 0.707 |
| Anterior View Shoulder Shift (cm) | PNF and Ex | −0.54 ± 1.61 | −0.46 ± 1.49 | −0.36 ± 1.53 | T: 0.606 | 0.483 | 0.017 |
| | Ex | −0.33 ± 1.84 | −0.36 ± 1.65 | −0.25 ± 1.47 | T*G: 0.236 | 0.694 | 0.007 |
| Anterior View Shoulder Tilt (cm) | PNF and Ex | −1.99 ± 0.72 | −1.34 ± 1.22 | −1.16 ± 1.30 | T: 6.547 | 0.010[b] | 0.144 |
| | Ex | −0.21 ± 3.14 | −0.24 ± 2.78 | −0.17 ± 2.86 | T*G: 6.959 | 0.008[b] | 0.151 |
| Lateral View Shoulder Shift (cm) | PNF and Ex | −0.69 ± 3.18 | −0.51 ± 2.68 | −0.57 ± 2.87 | T: 1.087 | 0.32 | 0.024 |
| | Ex | −0.71 ± 4.66 | −0.6 ± 4.28 | −0.32 ± 4.35 | T*G: 0.568 | 0.5 | 0.012 |
| Lateral View Vertical Displacement of Shoulder (degrees) | PNF and Ex | 3.03 ± 1.04 | 1.5 ± 0.76 | 1.77 ± 0.69 | T: 142.723 | 0.000[b] | 0.756 |
| | Ex | 3.47 ± 1.33 | 3.13 ± 1-4 | 3.21 ± 1.43 | T*G: 59.201 | 0.000[b] | 0.563 |
| Anterior View Thorax Shift (cm) | PNF and Ex | 0.08 ± 1.31 | 0.37 ± 1.26 | 0.32 ± 1.33 | T: 0.438 | 0.646 | 0.01 |
| | Ex | 0.22 ± 2.12 | 0.09 ± 1.93 | 0.28 ± 1.94 | T*G: 1.16 | 0.318 | 0.026 |

Notes.

Abbreviations: F, variance of the group means divided by mean of the within group variances; $\eta^2$, measure of effect size; cm, centimetres; PNF, proprioceptive neuromuscular facilitation; Ex, exercise; T, time; G, group.

[a]Values are mean ± SD.

[b]$p < 0.05$.

**Table 4 Comparison of Walking Speed, Timed Up and Go, Dynamic Gait Index and Functional Reach Test differences.**

| Outcome[a] | Group | Before treatment | After treatment | Follow up | F | P value | $\eta^2$ |
|---|---|---|---|---|---|---|---|
| Walking Speed (m/s) | PNF and Ex | 0.57 ± 0.20 | 0.65 ± 0.19 | 0.60 ± 0.20 | T: 6.062 | 0.003[b] | 0.116 |
| | Ex | 0.52 ± 0.20 | 0.55 ± 0.16 | 0.53 ± 0.14 | T*G: 1.154 | 0.320 | 0.024 |
| Timed Up and Go Test (s) | PNF and Ex | 13.34 ± 4.47 | 11.26 ± 3.71 | 12.13 ± 4.34 | T: 37.478 | 0.000[b] | 0.449 |
| | Ex | 15.88 ± 7.18 | 13.64 ± 5.45 | 14.28 ± 6.07 | T*G: 0.302 | 0.740 | 0.007 |
| Dynamic Gait Index | PNF and Ex | 16.96 ± 3.95 | 20.61 ± 2.71 | 19.70 ± 2.84 | T: 100.185 | 0.000[b] | 0.685 |
| | Ex | 15.68 ± 3.52 | 19.00 ± 3.51 | 18.44 ± 3.34 | T*G: 0.291 | 0.748 | 0.006 |
| Functional Reach Test (cm) | PNF and Ex | 22.26 ± 7.91 | 25.39 ± 7.21 | 24.63 ± 7.40 | T: 23.309 | 0.000[b] | 0.336 |
| | Ex | 19.94 ± 6 | 22.01 ± 6.03 | 20.93 ± 5.72 | T*G: 1.766 | 0.177 | 0.037 |

Notes.

Abbreviations: F, variance of the group means divided by mean of the within group variances; $\eta^2$, measure of effect size; m/s, meters per second; PNF, proprioceptive neuromuscular facilitation; Ex, exercise; T, time; G, group; s, seconds; cm, centimetres.

[a]Values are mean ± SD.

[b]$p < 0.05$.

*Hwang, Lee & Lim (2021)* examined scapular position, movement, and functions in 42 office workers with scapular dyskinesis. Patients were randomly allocated to three groups (muscle strengthening, muscle balance, and movement control). The movement control group underwent scapular PNF with the aim of improving dynamic stabilization of the
scapula three days a week for six weeks. The results of the study demonstrated that all outcome measurements improved in favor of the movement control group (*Hwang, Lee & Lim, 2021*). Similar to our study, the scapular position was corrected (*Hwang, Lee & Lim, 2021*). In our study, general posture, functionality, and balance were included in the outcome measurement as the included participants were older adults whose main difficulty was balance and postural problems that could be worsened by scapular dyskinesis. We observed statistically significant improvements in balance, posture, and postural outcomes.

According to the literature, scapular PNF interventions are effective in improving the LSST results (*Hwang, Lee & Lim, 2021*; *Joshi, Shridhar & Jayaram, 2020*). In parallel with the literature, our results showed that both groups had significant improvements in all three positions. When the two groups were compared, the PNF+Ex group showed statistically significant improvement. As scapular PNF is mainly effective for scapular muscle (serratus anterior, trapezius, rhomboids) strength and coordination, a significant improvement in the LSST results was expected. When follow-up measurements were observed, because the results tended to decrease, we estimated that the effects could be transient; however, the effectiveness was still significant. Therefore, we can conclude that scapular PNF interventions combined with exercise have more sustainable effects on scapular dyskinesis than balance and gait exercises alone do.

According to literature, PNF is one of the most effective interventions for improving posture (*Cho & Gong, 2017*). As postural control requires interactions between the nervous and musculoskeletal systems (*Sedaghati, Ahmadabadi & Goudarzian, 2022*), PNF helps improve this link. According to the results of our study, the lateral-view vertical displacement of the head, anterior-view shoulder tilt, and lateral-view vertical displacement of the shoulder significantly improved in the PNF+Ex group. In a total analysis of the postural results, normalizing the symmetry of the scapula with improving scapular muscle strength improved spinal alignment and postural stability, as the PNF interventions facilitated participants' neuromuscular control and scapular alignment (*Joshi, Shridhar & Jayaram, 2020*; *Sedaghati, Ahmadabadi & Goudarzian, 2022*). While improvements in scapular position and posture were promising, effect of scapular PNF intervention on functional mobility results needs further investigation.

For functional mobility and balance, as stated in the literature, decrease in balance and falls are directly related to changes in posture and functionality in the older adults (*Sedaghati, Ahmadabadi & Goudarzian, 2022*). The literature states that PNF combined with balance training significantly increases balance (*Kumar & George, 2024*; *Nguyen, Chou & Hsieh, 2022*). The results of this study showed that balance and functional mobility were significantly improved in both groups, but there was no significant difference between the groups. According to the data obtained in this study, after eight weeks of intervention, even though there was no significant difference between the groups, the observed postural improvements in the PNF+Ex group together with scapular alignment corrections might have evolved into being more effective on functional mobility and balance if the treatment time was extended (*Peteraitis & Smedes, 2020*). Therefore, including PNF to improve postural alignment and balance may be an effective approach. In addition to balance and

functional mobility, gait characteristics were also measured to determine the extended functional impact of the PNF intervention.

Previous studies have reported that PNF intervention is effective in improving gait characteristics (*Nguyen, Chou & Hsieh, 2022*). As PNF was effective in improving core stability, muscle strength, and posture (*Nguyen, Chou & Hsieh, 2022*), significant improvements in the DGI and walking speed results were expected. In the current study, significant improvements were found in both groups; however, there was no significant difference between the groups. The results of this study showed that although there was no significant difference between the groups, the increase in walking speed was higher in the PNF+Ex group. Despite these findings, it is important to consider some methodological limitations for this research.

Although mainly improvements in scapular position and posture were achieved, short treatment time (8 weeks) and follow-up (3 months) may have affected the functional improvements in balance and gait parameters. Additionally, the partial decline observed in scapular control over time suggests that longer treatment period and extended follow-up may be necessary to sustain and maximize improvements in older adults with scapular dyskinesis.

In summary, previous studies that observed the effects of scapular PNF in older adults with scapular dyskinesis had the main outcomes of muscle strength and scapular alignment; thus, their treatment time, which was mainly between 4 and 8 weeks, was sufficient, and statistically significant improvements were observed (*Bhagat, Harishchandre & Ganvir, 2023*; *Ciğercioğlu et al., 2022*; *Hwang, Lee & Lim, 2021*). In the current study, treatment time was initially adapted from previous studies, and more significant differences in functional level could be obtained together with longer treatment time (*Lee, Hwangbo & Lee, 2014*; *Shaikh & Ganvir, 2023*).

## Limitations

The same therapist conducted both the training and assessments. Although blinding was not feasible, objective measurement tools were used to minimize subjective bias. We recommend that future studies include blinded assessments to improve validity. Another limitation of the study was the lack of long-term follow-up and the inability to monitor the duration of the improvements, which we believe will be more relevant in long-term rehabilitation. In addition, the evaluations were not performed at the standard evaluation time of the day. In older age, we estimate that the fatigue of the participants evaluated in the evening hours may negatively affect the test results compared with the participants that were evaluated in the morning hours. Regarding measurement methods, during LSST participants could alter their movements and attempt to stand up straight during the measurement. For future studies as part of outcome measurements, we suggest that muscle strength measurements should be included, especially trunk and upper extremity muscle strength, to estimate the effects on balance and gait, and objective posture and gait analysis methods should be included in the measurements, especially for gait parameters, and arm swing data should have been obtained. Lastly, future studies should include longer

follow-up periods to determine the long-term impact of scapular PNF interventions in older adults.

## CONCLUSIONS

Our study showed that the implementation of scapular PNF in the rehabilitation of older adults with scapular dyskinesis is beneficial for enhancing gait characteristics, balance, and posture. Although conventional physiotherapy is known to be a successful rehabilitation method for older adults with thoracic kyphosis accompanied by scapular dyskinesis, improving muscular activity around the scapula by including scapular PNF intervention has more advantages than conventional exercise therapies. We suggest that scapular PNF exercises are an effective and safe rehabilitation method for older adults with scapular dyskinesis.

## ACKNOWLEDGEMENTS

We would like to extend our sincere gratitude to all the participants who contributed their time and effort to this study.

### Funding

The authors received no funding for this work.

### Competing Interests

The authors declare there are no competing interests.

### Author Contributions

- Hasan Cellatoğlu conceived and designed the experiments, performed the experiments, analyzed the data, prepared figures and/or tables, authored or reviewed drafts of the article, and approved the final draft.
- Beliz Belgen Kaygısız conceived and designed the experiments, prepared figures and/or tables, authored or reviewed drafts of the article, and approved the final draft.

### Ethics

The following information was supplied relating to ethical approvals (*i.e.*, approving body and any reference numbers):

This research was approved by the University Ethics Committee of the University of Lefke (BAYEK022.04).

### Data Availability

The raw data is available in the Supplementary Files.

### Clinical Trial Registration

The following information was supplied regarding Clinical Trial registration:

NCT05893303

## Supplemental Information

Supplemental information for this article can be found online at http://dx.doi.org/10.7717/peerj.19718#supplemental-information.

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
