# Peer review of "Effect of proprioceptive neuromuscular facilitation (PNF) technique on posture, balance and gait characteristics of older adults with scapular dyskinesis: a randomized controlled trial"

_PeerJ, doi:10.7717/peerj.19718_

## Round 0.1 · original submission · Major Revisions

Reviewers found merit in your manuscript but have offered suggestions for improvement to the flow and clarity in certain places. Please carefully consider and implement feedback from both reviewers.

Reviewer 1 ·

Basic reporting

The article is written in clear and concise English. Line 182 uses the British English spelling of "labelled" would recommend switching to the American English spelling of "labeled".

The authors do a good job of connecting PNF with rehab of scapular dyskinesis and posture. However as a reader I am not able to draw as clear of a connection between scapular dyskinesis and improving gait and balance. I understand that the author is attempting to connect functional limitations in line 108 with potential disturbances in gait and balance, however I would like to see the authors include more support from existing literature to help the reader follow the the thought process here. The authors point to limited literature, but maybe they can expand to help reviewers and potential readers. Are balance & gait only effected when a subject has combined dyskinesis & thoracic kyphosis? The study itself focuses only on treatment of dyskinesis.

With that said, please clarify the connections between dyskinesis, thoracic kyphosis, balance, gait and posture.

Experimental design

no comment

Validity of the findings

conclusions appear to be vaild. The discussion is thorough, however it appears to lack some flow, clarity and understanding. The authors should consider cleaning this section up to increase clarity and understanding of future directives.

Additional comments

Table 1 fails to provide information on male subjects. Please provide clarity on male subjects.

While the authors do extensive research on their topic more clarity is needed on scapular dyskinesis with and without kyphosis and how this may effect the relationships that the authors are exploring.

·

Basic reporting

The paper reads smoothly, with professional language that’s easy to follow for a global audience. The introduction lays out the basics of scapular dyskinesis and PNF’s role in rehab, citing studies like Hwang et al. (2021) and Joshi et al. (2020). It’s organized logically, with sections for methods, results, and discussion that align with PeerJ’s guidelines. The flow diagram (Figure 1) and tables (1–4) neatly summarize participant data, posture, and functional results. Sharing raw data is a big plus for openness.

Areas for Improvement:
The background could use a couple of newer references to show the latest on PNF for older adults. Figure 1 needs a clearer caption to explain the participant flow. There are small typos, like "cantimeters" instead of "centimeters" in Table 2, and abbreviations aren’t always consistent. Also, the PostureScreen Mobile (PSM) app is introduced without explaining why it’s reliable, which might confuse some readers.

Suggestions:
Add 1–2 recent studies to the introduction.
Write a detailed caption for Figure 1.
Fix typos and make abbreviations consistent (e.g., always use "cm").
Mention PSM’s reliability early on, citing Boland et al. (2016).

Experimental design

This is original work that fits PeerJ’s focus, exploring how PNF helps older adults with scapular dyskinesis—a topic with limited prior research. The question is clear: does PNF plus exercise beat exercise alone for posture, balance, and gait? The study’s ethical setup is solid, with approval from the University of Lefke (BAYEK022.04), ClinicalTrials.gov registration (NCT05893303), and informed consent. The methods section covers interventions, measures (LSST, PSM, TUG, DGI, FRT), and stats (ANOVA) in enough detail to repeat the study.

Areas for Improvement:
Not blinding the therapist who did both assessments and treatments is a weak point, as it risks bias. The authors mention this but don’t dig into how they managed it. The sample size is based on a study about elderly women’s balance (Mesquita et al., 2015), which feels like a stretch for this context. It’s also unclear if they hid group assignments during randomization to avoid bias. Finally, the exercise and PNF descriptions could be more specific about intensity or progression.

Suggestions:
Explain how objective measures reduced bias from no blinding.
Use a more relevant study for sample size or justify the choice better.
Confirm if group assignments were concealed during randomization.
Add details on exercise intensity (e.g., reps, resistance) for clarity.

Validity of the findings

The data are complete and analyzed well, using ANOVA and chi-square tests, with normal distribution checked via Shapiro-Wilk. The sample size was calculated for 95% power, and effect sizes in Tables 2–4 make the results easy to interpret. The conclusions stick to what the data show, highlighting PNF’s benefits for posture and scapular alignment while suggesting more time for gait/balance gains. The findings build on past work (e.g., Hwang et al., 2021) by focusing on older adults.

Areas for Improvement:
The lack of group differences in balance/gait measures (TUG, DGI, FRT, walking speed) undercuts the idea that PNF is better. The discussion could explore why—maybe the 8-week timeline was too short? The 3-month follow-up is brief, and LSST improvements faded, hinting at short-lived effects. Some stats, like F-values (e.g., "F=5,414"), are formatted oddly and could confuse readers. Raw data files need better labels to be useful for others.

Suggestions:
Discuss possible reasons for no balance/gait differences (e.g., study duration, measure sensitivity).
Fix F-value formatting (e.g., "5,414" to "5.414" or "5414").
Add clear labels to raw data (e.g., define variables, units).
Suggest studies with longer follow-ups to check lasting effects.

Additional comments

General Comments:
This study offers practical insights for rehab specialists, showing PNF’s value for posture in older adults with scapular dyskinesis. The randomized design and range of measures (LSST, PSM, TUG, DGI, FRT) make it robust. The authors handled ethics well and shared data openly. That said, no blinding and a short follow-up are limitations, and small language/formatting issues need tidying up.

---

## Round 0.2 · Minor Revisions

Reviewer 1 has provided some small things that need to be addressed. Thank you to the authors for their patience, I know this took a bit longer than expected.

Reviewer 1 ·

Basic reporting

Basic reporting is clear and authors have worked to increase the flow of the paper, therefore increasing both the readability and clarity of the paper.

Authors increased the lit review to show the connection between scapular dyskinesis and thoracic kyphosis, as well as thoracic kyphosis and the broader impact on postural control. This helps the reader to draw conclusions between the primary purpose or hypothesis of "purpose of this study was to analyze whether including scapular PNF in an intervention improves the posture, balance, and gait characteristics of older adults with scapular dyskinesis" and the methods.

Some minor flaws in the writing remain and will be addressed below line by line for assistance with revision.

Lines 67-70 there is a typing error. Please remove the period after dysfunction and prior to the (references) in line 69 and move the references to the end of the sentence in line 70 as this appears to be a continual sentence and lends to the connection between scapular dyskinesis and thoracic kyphosis and therefore postural control.

Lines 170-172 appear to repeat themselves with partial discrepancies/contradictions. Line 170-171 states that VSDT and LSST were performed to detect the presence of scapular dyskinesis. Lines 171-172 states that to detect scapular dyskinesis a VSDT was conducted. Please clarify this. maybe the authors should consider saying " To detect the presence of scapular dyskinesis a VSDT was performed. This would add clarity as in line 174 you state the purpose of LSST.

Lines 180 & 181 Could be reversed for clarity and flow. "Posture measurements were performed utilizing the PostureScreen Mobile App (PSM) (PostureCo Inc). According to Boland et al (2016), PSM is a validated and reliable tool for postural assessment." Starting the paragraph/ section with a claim about the validity before introducing it to the reader as the device utilized for assessing posture decreases the flow and readability of this paragraph.

Lines 183-187 I would also encourage the authors to spell out the app usage. Were the pictures mentioned in line 183 taken by the researchers and uploaded to the app, were the pictures taken in a step by step process directed in the app, or were pictures provided by the subject. This helps to gain clarity and the ability to reproduce in future studies.

Lines 195-200 The authors cite a paper to show the reliability of the test which is very helpful, but it may increase reader comprehension if you would describe the test in slightly more detail to gain a full understanding.

Lines 201-207 Similarly it would be useful for the authors to describe the specific tasks to increase awareness, understanding and reproduce-ability of the test.

Line 324 recommend removing the line "Because this was a PhD thesis declaration.

Experimental design

no comments

Validity of the findings

no comments

Additional comments

The authors have noticeably improved the paper at this time. They have increased clarity and readability of the paper. They increase the understanding of the connections between the methods and main purpose stated in the abstract. However some necessary clarifications remain needed at this time and minor revisions are suggested.

·

Basic reporting

The manuscript is written in professional and clear English.
The manuscript includes a well-structured literature review, citing relevant studies to support the research objectives.
The paper follows a professional structure consistent with academic publishing standards.
The authors have provided raw data files, ensuring transparency in their methodology.
The manuscript is self-contained and effectively ties its results back to the hypotheses.

Experimental design

The study falls within the journal’s scope, addressing rehabilitation techniques for older adults with scapular dyskinesis through a randomized controlled trial.
The authors have clearly formulated their research question, investigating whether incorporating scapular PNF improves posture, balance, and gait characteristics compared to conventional exercise. The study fills an identified knowledge gap, as few prior studies have specifically examined scapular PNF in this context.
The study adheres to ethical guidelines, including approval from the University Ethics Committee. The inclusion/exclusion criteria for participants are clearly stated, ensuring a controlled and focused sample group.
The intervention is well-documented, detailing the duration, frequency, and specific exercises. However, the manuscript could enhance replicability by adding finer details on exercise intensity and progression strategies for both groups.

Validity of the findings

The study is well-structured and clearly outlines its research gap.
The study follows a solid statistical framework, using appropriate methods (ANOVA, chi-square tests) to compare intervention effects.
The conclusions are coherent and directly connected to the study’s hypothesis.

Additional comments

The study is comprehensive and methodologically sound, contributing valuable insights into the role of scapular PNF in rehabilitation.

---

## Round 0.3 · accepted · Accept

Comments have been addressed.

Reviewer 1 ·

Basic reporting

All basic reporting is clear, concise and appears to meet Peer J standards at this time.

Experimental design

Appears to meet Peer J standards at this time.

Validity of the findings

Appears to meet Peer J standards at this time.

Additional comments

Authors have addressed the final requests for revision and the paper appears to meet PeerJ publication standards at this time. Thank you for allowing me to review your manuscript and contribute to the profession in a positive manner.